# Formation Pathways of Lath-Shaped WO_3_ Nanosheets and Elemental W Nanoparticles from Heating of WO_3_ Nanocrystals Studied via In Situ TEM

**DOI:** 10.3390/ma16031291

**Published:** 2023-02-02

**Authors:** Xiaodan Chen, Marijn A. van Huis

**Affiliations:** 1Soft Condensed Matter, Debye Institute for Nanomaterials Science, Utrecht University, Princetonplein 5, 3584 CC Utrecht, The Netherlands; 2Electron Microscopy Center, Utrecht University, Universiteitsweg 99, 3584 CG Utrecht, The Netherlands

**Keywords:** tungsten trioxide, phase transformation, in situ electron microscopy, nanosheets

## Abstract

WO_3_ is a versatile material occurring in many polymorphs, and is used in nanostructured form in many applications, including photocatalysis, gas sensing, and energy storage. We investigated the thermal evolution of cubic-phase nanocrystals with a size range of 5–25 nm by means of in situ heating in the transmission electron microscope (TEM), and found distinct pathways for the formation of either 2D WO_3_ nanosheets or elemental W nanoparticles, depending on the initial concentration of deposited WO_3_ nanoparticles. These pristine particles were stable up to 600 °C, after which coalescence and fusion of the nanocrystals were observed. Typically, the nanocrystals transformed into faceted nanocrystals of elemental body-centered-cubic W after annealing to 900 °C. However, in areas where the concentration of dropcast WO_3_ nanoparticles was high, at a temperature of 900 °C, considerably larger lath-shaped nanosheets (extending for hundreds of nanometers in length and up to 100 nm in width) were formed that are concluded to be in monoclinic WO_3_ or WO_2_._7_ phases. These lath-shaped 2D particles, which often curled up from their sides into folded 2D nanosheets, are most likely formed from the smaller nanoparticles through a solid–vapor–solid growth mechanism. The findings of the in situ experiments were confirmed by ex situ experiments performed in a high-vacuum chamber.

## 1. Introduction

Tungsten trioxide is a semiconductor material with very diverse chemical and physical properties, and is consequently used in very diverse applications, including photocatalysis [1,2,3,4], gas sensing [5,6,7,8], energy storage [9,10,11,12], and as an electrochromic [13,14] material. WO_3_ is widely applied as it is available at low cost, is abundant, and has an open tunnel-like structure, which makes it permeable to gas atoms and suitable for ion transport. 

The morphology and crystal structure are strongly connected to the electronic properties of nanostructured WO_3_ [14,15,16], and consequently affect their applications in catalysis and energy storage. Furthermore, gas sensors are expected to function as well in high-temperature environments and therefore, an in-depth understanding of the thermal behavior and thermal stability of nanostructured WO_3_ is of vital importance to assess their applicability to high-temperature applications. 

There are various crystalline polymorphs of WO_3_, which are based on a cubic ReO_3_ structure [17]. The material consists of tungsten-centered oxygen octahedrons (WO_6_ octahedrons) that are corner-sharing and that show distortions, forming different phases with lower symmetry. Figure 1 shows the structure of the main four polymorphs with oxygen octahedrons. Corresponding crystallographic information, including space groups and lattice parameters, are listed in Appendix A. W atoms are at the center of every octahedron. In previous studies, phase transformations between different polymorphs were observed in many cases, as a result of temperature treatment [18,19,20,21,22], doping [23,24,25,26,27,28,29], or mechanical treatment [30]. The most common stable phase at room temperature is the monoclinic crystal structure (Space group P2_1_/*n*). With increasing temperature, the most stable phases are orthorhombic (P*bcn*, ~500 °C), tetragonal (P4/*ncc*, 800 °C), and tetragonal (P4/*nmm*, 900 °C) [17,19,20,31]. Furthermore, there are metastable phases, such as hexagonal, triclinic, and cubic phases. In an investigation by Howard et al. [32], another monoclinic phase (P2_1_/*c*) was observed to be formed between 760 °C and 800 °C. It was found by Han et al. that this monoclinic phase can co-exist with the tetragonal phase under certain conditions [33]. Ramana et al. [18] reported that monoclinic WO_3_ thin films transformed into the hexagonal phase at 500 °C. The thermal behavior of some metastable phases was also investigated [21,22]. However, few studies have investigated the highest symmetry cubic WO_3_ phases that are investigated in the present work.

Cubic WO_3_ (P*m*$\bar{3}$*m*) is not a stable phase reported in the W–O phase diagram [17]. Corà et al. explained the reason for its instability as bulk material in 1996 from Hartree–Fock calculations [34]. Nevertheless, nanosized cubic-phase WO_3_ has been successfully fabricated [35,36], is commercially available, and has been used for solar cells [10] and as an anode material [11]. The aim of this work is to assess the thermal stability and to characterize occurring phase transformations and morphology changes of cubic WO_3_ phase nanoparticles. The thermal evolution of the nanocrystals is investigated from room temperature to 1000 °C with in situ heating transmission electron microscopy (TEM) in order to study their structural and chemical thermal evolution in detail and in real time [37]. Most particles transformed into pure cubic α-W at 900 °C. At the same temperature, bigger lath-shaped WO_3_ nanosheets were formed by recrystallization into a monoclinic structure. Transmission electron microscopy (TEM), selected area electron diffraction (SAED), and 2D chemical mapping via electron-dispersive X-ray spectroscopy (EDS) were employed for phase identification and to monitor structural and chemical transitions.

## 2. Experimental

The WO_3_ nanocrystals (NCs) were purchased from Sigma-Aldrich (Product Number: 807753). All in situ TEM investigations and STEM-EDS measurements were conducted using a TFS TalosF200X TEM operating at 200 kV. The high-resolution (HR) STEM images were taken with a double aberration-corrected TFS Spectra300 TEM operating at 300 kV. The specimens were prepared by drop-casting the WO_3_ nanoparticle solution onto a DENSsolutions MEMS heating chip.

The heating chips were subsequently mounted onto a DENS Solutions single-tilt heating holder. The WO_3_ nanoparticles were first heated from 20 °C to 1000 °C in 100 °C increments. The nanoscale phase transformation happened at 900 °C. In a second heating experiment, the specimen was heated from 20 °C to 800 °C in 100 °C increments, then in smaller increments of 25 °C when raising the temperature further from 800 °C to 900 °C, in order to monitor the possible presence of intermediate phases. The particles were found to be sensitive to the electron beam at elevated temperatures. Appendix A shows that the particles deformed rapidly after illumination by the electron beam for 1 min. In order to avoid such electron beam effects, the field of view was changed very often in order to always examine an area that was not previously exposed to the electron beam (the electron beam illuminates only a tiny fraction of the sample deposited area). Furthermore, in order to fully exclude any electron beam effects, the samples were also heated ex situ outside of the TEM. For these ex situ experiments, the samples were heated with the heating holder inserted in a high-vacuum chamber (Gatan pumping station Model 655), applying the same heating rate as in the in situ heating experiments. The pressure in the high-vacuum chamber was approximately 1.0·10^−7^. Torr. After holding the temperature at 900 °C for 10 min, the sample was cooled down fast to room temperature and swiftly inserted in the TEM for subsequent analysis.

## 3. Results and Discussion

Figure 2 shows the overview (a) and high-resolution (b) bright-field TEM images of the as-received WO_3_ specimen at room temperature. The nanoparticles have a broad size range of 5–25 nm. Both the lattice fringes in the high-resolution TEM image in (b) and the selected-area diffraction pattern (SADP) with indexed diffraction rings in (c) confirm the cubic crystal structure.

The WO_3_ nanoparticles were heated from room temperature to 1000 °C in 100 °C increments. Figure 3 shows bright-field TEM images of the specimen heated at different temperatures, displaying the evolution in morphology during heating. The images were taken from different areas of the heating chip in order to prevent any influence of the electron beam illumination on the observation of the thermal evolution, as explained in the Experimental section. Up to 600 °C, there was no obvious deformation of the particles. At 700 °C, the particles began to coalesce. At the edges of the particle clusters, some particles sublimated and left smaller dots. At 800 °C, coalescence progressed, and small dots appeared commonly around the original particles. After annealing at 900 °C, the particles lost their original shape completely. In some areas, big lath-shaped particles were formed, as can be seen in the bottom-right image from Figure 3. The lath-shaped particles were sensitive to the beam at high magnification, similarly to the e-beam sensitivity of the nanoparticles.

The SADPs were used for tracking and identifying the phase changes during heating. As shown in Appendix A, a phase transformation took place at 900 °C. The diffraction pattern was indexed and is shown in Figure 4c, which indicates that the resulting phase was pure, body-centered cubic (bcc) W (α-W). High-resolution (HR) TEM images recorded along different projections of the crystal structure also confirmed the cubic W crystal structure. Therefore, the cubic-phase WO_3_ nanoparticles transformed into cubic-phase W nanoparticles at 900 °C.

There are many other tungsten oxides with lower oxidation state and a composition between WO_3_ and pure W. The formation of other tungsten oxides, such as WO_2_._9_, WO_2_._72_, and WO_2_, has been observed during the reduction of WO_3_ by many researchers [38,39,40,41,42,43]. We conducted additional experiments to check for the presence of any intermediate phases before pure W was formed. The specimens were heated to 800 °C in 100 °C increments, and subsequently from 800 °C to 900 °C in smaller increments of 25 °C. Appendix A shows the SADPs from 800 °C to 875 °C, where no other rings appeared in any of these DPs. This means that we did not observe any other intermediate phases. One noteworthy observation is that the intensity of the third ring (marked with a blue arrow) increased gradually from 825 °C onward (shown in Appendix A). This ring corresponds to both the (111) lattice reflection of WO_3_ and the (110) lattice reflection of W. Therefore, cubic W is possibly already formed slightly below 900 °C. 

The observed direct transformation to pure W, which is different from previous studies, could be attributed to the high heating rate in our case. In the research of Fouad et al. [43], both isothermal and non-isothermal TGA measurements were taken. During the non-isothermal mode measurements, samples were heated up to 1000 °C at a rate of 10 °C/min. Three transformation steps happened at 520–600 °C (WO_2_._7_), 600–655 °C (WO_2_), and 713–875 °C (W). However, during isothermal measurement, the samples were kept at constant temperature. When their powder sample was measured at 740 °C, the intermediate transformations overlapped kinetically. Only one steep step was detected in that study, corresponding to the complete reduction of WO_3_ to W. In our study, the heating rate was considerably higher than 10 °C/min, implying that, in our case, several transformation steps would be overlapping, resulting in direct transformation to pure W. We mention here that Fouad and co-workers performed their study on WO_3_ powder initially having a monoclinic crystal structure, while the present study was conducted on smaller WO_3_ nanoparticles that initially had a cubic crystal structure, which explains the different thermal evolution observed in the present investigation. Until now, there have been very few investigations reported in the literature on the thermal stability and reduction of cubic-phase WO_3_, and therefore, follow-up investigations using complementary methods, such as in situ XRD and TGA/DSC conducted on cubic-phase WO_3_, would be interesting to obtain further insights into the observed processes.

To rule out any possible influence of e-beam illumination to the phase transformation, ex situ heating experiments were conducted in a vacuum chamber outside the microscope, where the particles were heated to 900 °C as well, after which they were inserted in the TEM for structural characterization. Surprisingly, in one of the experiments where a large amount of particles was dropcast onto the heating chip, many large lath-shaped particles were formed, and the DP also showed a strong peak of WO_3_ (shown in Figure 5a,b). However, when fewer particles were dropcast onto the heating chip, lath-shaped particles were not formed, and the DP only indicated the cubic W crystal structure as shown in Figure 5c,d. It seems that, when the concentration of WO_3_ nanoparticles is sufficiently high, lath-shaped particles can be formed, and this formation of lath-shaped particles apparently impedes the transformation to cubic W.

In many images, the shape of the lath-shaped particles resembled that of a rod where the varying contrast points to curling up of the laths into cylinder-like structures. To investigate the shape of these particles further, the sample was tilted to approximately ±30° along the α-tilt axis. Appendix A shows the images of an area projected along two tilt angles. The width of the particles changed by tilting, implying that the particles are not perfectly cylindrical. The particle shown in Figure 6 looks like a sheet that is folded at two sides, like the model shown in panel (c). The STEM images also show less contrast at the center, indicating lesser thickness. Therefore, in this paper, we named the larger particles lath-shaped nanosheets. 

Figure 6e,f shows magnified TEM images of the corresponding areas in panel (a). The spacings of the fringes on the two sides of the particles are much bigger than the lattice spacings. These stripe-like moiré patterns occur when two or more layers are overlapping while having different lattice spacings in projection, and the lattice fringes are aligned in the same direction. The spacing of the moiré fringes *d_tm_* formed by the different projected lattice spacings *d*_1_, *d*_2_, can be calculated using the following equation:$$d_{tm}=\frac{d_{1}}{1-\frac{d_{2}}{d_{1}}}$$

The moiré patterns visible in Figure 6e,f were commonly observed on the lath-shaped particles. Additional similar images are shown in Appendix A. The moiré patterns indicate multiple layers, confirming that the two sides of the nanosheets have folded edges.

Figure 7 shows HR images and STEM-EDS chemical mapping of the lath-shaped particles. From the HR images, different lattice spacings are detected along the length and width of the particles, indicating that the particles are not cubic anymore. The lattice spacings along the length of the particles are about 3.85 Å, which corresponds to the (002)-plane of the P2_1_/*n* monoclinic structure. The lattice spacings in the lateral directions in panels (a) and (b) are 3.62 Å and 3.76 Å, respectively, corresponding to the (200) and (020)-spacings of the monoclinic phase. Therefore, these lath-shaped particles grow in length along the monoclinic c-axis. 

In the research of Pokhrel et al. [20], the monoclinic nanocrystals were heated and several phase transformations were detected during heating, where elongated particles were observed between 800 and 950 °C. Their XRD results showed a tetragonal phase in this temperature range. Consequently, these authors concluded that the elongated particles had the tetragonal structure. However, in our case, in the DP of the lath-shaped particles (shown as Figure 5b), there are peaks corresponding to a lattice spacing of 4.2 Å, which do not belong to the tetragonal phase. Moreover, the lath shape of the particles indicates that the growth rate along the three crystal axes is distinctly different, which agrees better with the monoclinic phase than with the tetragonal phase.

There are also HR images showing lath-shaped particles with a structure that differs from that of monoclinic WO_3_. In Figure 7c, the lattice spacing along the length of the lath-shaped particle is 3.86 Å, while the lateral lattice spacing is 3.0 Å, which does not correspond to any of the projections of monoclinic WO_3_, but which matches very well the [106] projection of WO_2_._72_ (space group P2/m). WO_2_._72_ also has monoclinic structure, and was often observed as the intermediate phase during the reduction of WO_3_ [38,43]. Due to the presence of oxygen vacancies, the corner-sharing oxygen octahedra matrix of WO_3_ is distorted and, as a result, decahedra also exist in the WO_2_._72_ structure (shown in Appendix A). The 3.86 Å lattice spacing corresponds to the (010) interplanar spacing, and the lateral lattice direction is [$\bar{1}$06]. The distortion of the WO_6_ octahedra decreases the symmetry of the structure and results in many more lattice planes parallel to the b-axis than in the case of the monoclinic WO_3_ structure. The values of lateral lattice spacings in panels (a) and (b) also correspond to the (103) and (104) interplanar spacings of WO_2_._72_, respectively. This means that the particles in panels (a) and (b) could possibly be WO_2_._72_ when considering only lattice spacings. However, in the [$\bar{1}$03] and [$\bar{1}$04] projections of WO_2_._72_ (schematic structure shown in Appendix A), the lattices parallel to the b-axis are so condensed that the atomic columns would not be observed as sharp dots like those in the HR images shown in panels (a) and (b). Therefore, we can confirm that the structure shown in panel (c) is monoclinic WO_2_._72_, which means that WO_3_ and WO_2_._72_ lath-shaped particles co-exist. The lattice parameters of the WO_2_._72_ phase are listed in Appendix A. We mention here that the high-vacuum environment of both the TEM and the ex situ heating chambers is likely of importance for the formation of the WO_2_._72_ phase, as a very low partial oxygen pressure affects the relative stability of phases in favor of oxygen-deficient or oxygen-depleted phases, as explained in our previous work [44].

Figure 7d shows the STEM-EDS chemical mapping result. The mapping area includes both lath-shaped WO_3_ particles and small elemental W particles after heating. From the small particles that transformed into W, mainly an EDS signal of W was detected, proving that these particles were elemental W. The O signal on the W particles was due to surface reoxidation, since the EDS mapping was conducted a few weeks after heating. In contrast, the oxygen signal on the lath particle was stronger than the tungsten signal. The quantified chemical mapping resulted in a W:O ratio for the lath-shaped particles of 1:3.08, which within experimental errors agrees well with the WO_3_ or WO_2_._72_ composition of the lath-shaped particles.

In situ heating experiments were repeated to track the formation of the lath-shaped particles. Figure 8 shows images before and after the initial formation of a lath-shaped particle at 900 °C. On the left-hand side of the image, the future position of the lath-shaped particle is marked by a red rectangle, in which some areas are empty before the formation. Several of the surrounding particles (marked by the yellow arrows) disappeared after the formation of the lath-shaped particle. Unfortunately, as mentioned before, the specimens are very beam-sensitive at high temperatures. Therefore, the full formation process could not be recorded. 

The fabrication routes of 2D WO_3_ nanosheets or platelets were reported with various methods, including anodization [8], hydrothermal treatment [3], mechanical exfoliation [7], oxidation of WS_2_ [45], and colloidal chemistry methods [46]. In these reports, the synthesis took place mostly in a solution involving other chemical compounds, i.e., the synthesis routes were chemical rather than physical. WO_3_ and non-stoichiometric WO_3−x_ nanorods and nanowires can be fabricated with physical or chemical vapor deposition techniques [47,48,49,50,51,52,53,54,55,56]. Baek et al. [51] fabricated monoclinic WO_3_ nanowires on W substrate by heating WO_3_ powder under vacuum conditions. In the research of Hong et al. [53], WO_2_._72_ nanowires were synthesized via thermal evaporation of WO_3_ powder in vacuum, which is similar to the experimental conditions in our study. Zhang et al. [52] synthesized WO_2_._72_ nanowires on carbon microfibers by heating a W film in an atmosphere of Ar and water, and the WO_2_._72_ nanowires transformed into monoclinic WO_3_ after annealing at 500 °C. The growth of nanowires synthesized in these methods followed a vapor–solid mechanism, which likely also plays a role in the formation of lath-shaped particles in the current study. The vapor-solid mechanism is a common approach for forming nanostructures [57]. We hypothesize that, in our case, the WO_3_ nanocrystals started to sublimate at a temperature of 800 °C, and recrystallized very locally before the oxygen could disappear in the vacuum of the TEM column, corresponding to a solid–vapor–solid growth mechanism. Therefore, when a low concentration of pristine cubic-phase particles was deposited on the heating chip, the sublimated O atoms were pumped out of the column, resulting in cubic elemental W. From the results of the ex situ experiments displayed in Figure 5, it became clear that two distinct pathways can be selected by varying the concentration of deposited pristine particles: one pathway leading to the exclusive formation of elemental W nanoparticles, and one pathway leading to the predominant formation of 2D nanosheets of WO_3−x_.

## 4. Conclusions

The phase transformations and morphological changes of cubic-phase WO_3_ nanocrystals were investigated by in situ heating in the TEM. The initial particles were stable up to 600 °C, and began to coalesce and sublimate at 700 °C. Upon heating to 900 °C, most of the particles transformed into pure cubic-phase tungsten. Others coalesced and formed larger lath-shaped particles in the areas where the concentration of dropcast WO_3_ NPs was high. The lath-shaped particles were found to have monoclinic WO_3_ and WO_2_._72_ crystal structures. Sometimes, the lath-shape particles curled up from the sides, like folded 2D nanosheets. As also confirmed by the ex situ experiments, the heating of a low concentration of WO_3_ nanoparticles leads to the exclusive formation of elemental W nanoparticles, while the heating of a high concentration of WO_3_ nanoparticles leads to the predominant formation of lath-shaped WO_3−x_ nanosheets that are hundreds of nanometers long and up to ~100 nm wide, where the nanosheets are so thin that they are often found to curl up at their edges to form semi-cylindrical structures. The 2D character and the lath shape of the nanosheets are the result of their monoclinic crystal structure, which results in different growth rates along the three crystal directions. 

The current study has given detailed insights into the thermal stability of nanosized WO_3_ particles having a cubic crystal structure. We hypothesize that the lath-shaped particles with monoclinic crystal structure are formed through a solid–vapor–solid (SVS) growth mechanism. These 2D lath-shaped structures may be of particular use as catalytic or anode materials having a high surface area. It would be interesting to explore the functional properties of these spatially more extended lath-shaped structures in future studies.

## Figures and Tables

**Figure 1 materials-16-01291-f001:**
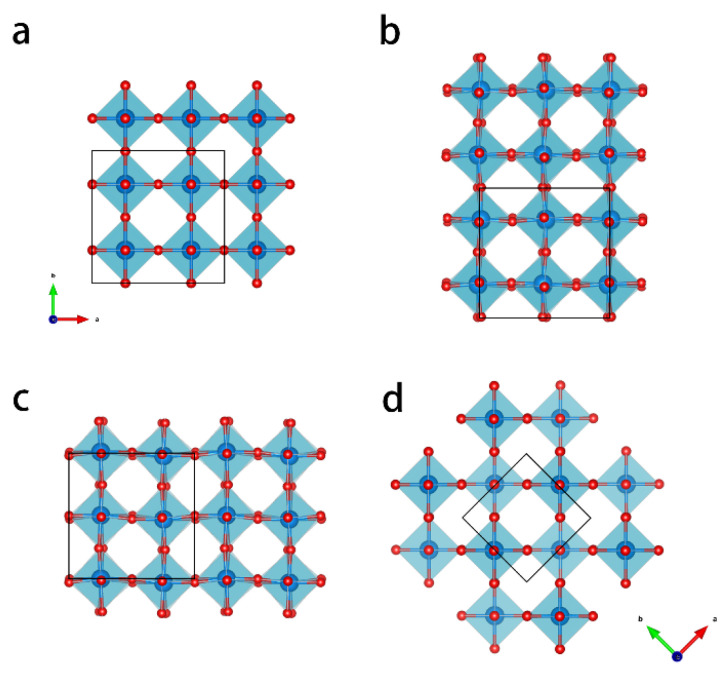
Crystal structures of the most commonly occurring WO_3_ polymorphs displayed in a [001] projection: (**a**) cubic; (**b**) room-temperature stable monoclinic; (**c**) orthorhombic; (**d**) tetragonal. The tetragonal phase has 45° tilt with respect to other phases. The unit cells are indicated with black lines. Crystallographic details are provided in Appendix A.

**Figure 2 materials-16-01291-f002:**
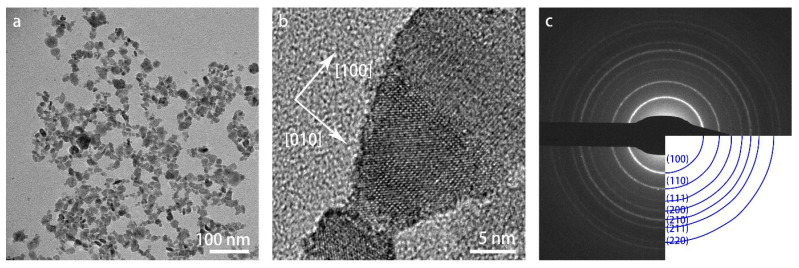
TEM images of WO_3_ nanoparticles at room temperature: (**a**) overview image in bright-field mode; (**b**) high-resolution image; (**c**) SAED pattern with indexing of the diffraction rings.

**Figure 3 materials-16-01291-f003:**
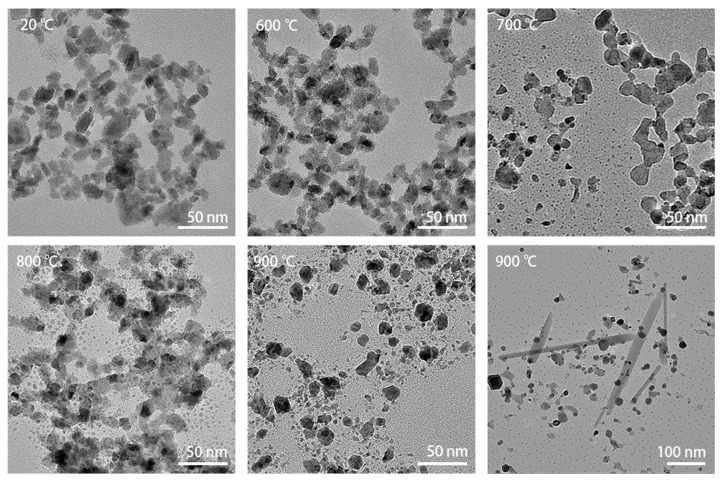
TEM images recorded during heating from 20 °C to 900 °C in steps of 100 °C. Up to 600 °C, there was no obvious deformation of the particles yet. At 700 °C, the particles began to coalesce, while some of the particles sublimated, leaving smaller dots of material behind. At 800 °C, coalescence continued and small dots commonly appeared around the original particles. After annealing at 900 °C, the particles lost their original shape completely. In some areas, much larger lath-shaped particles were formed as well, as can be seen in the bottom-right panel.

**Figure 4 materials-16-01291-f004:**
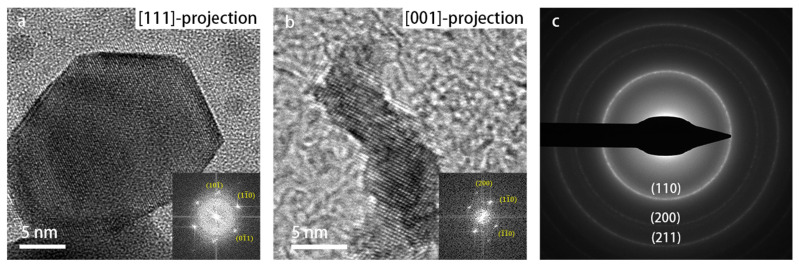
TEM images recorded at 900 °C: (**a**) HR-TEM image of elemental W in a [111]-projection; (**b**) HR-TEM image of W in a [001]-projection; (**c**) SADP at 900 °C, with the corresponding lattice reflections indexed.

**Figure 5 materials-16-01291-f005:**
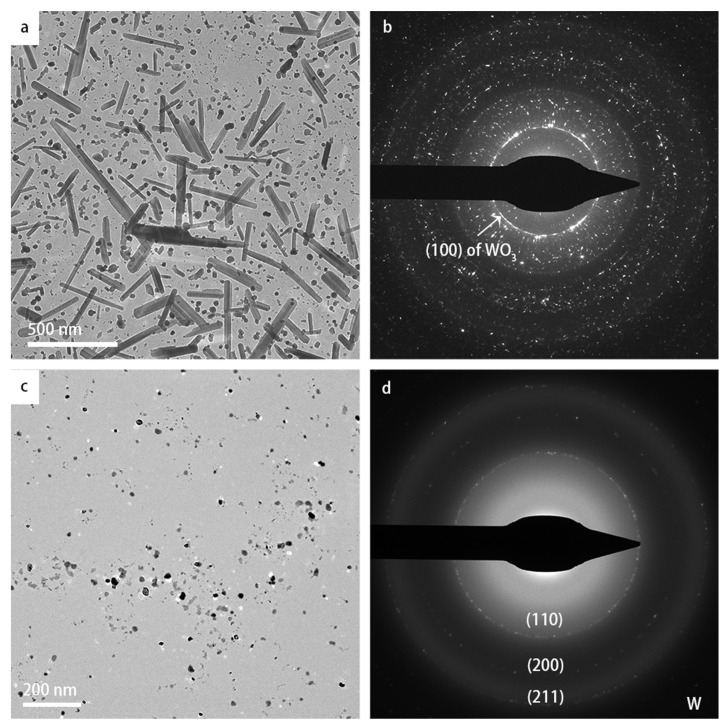
TEM images showing the results of the ex situ heating experiments: (**a**,**b**) TEM image and corresponding DP with high concentration of dropcast specimen; (**c**,**d**) TEM image and DP, respectively, with low concentration of dropcast specimen. Using the heating holder in a vacuum chamber, both samples were heated to a temperature of 900 °C with the same heating rate as in the in situ experiments. After keeping the temperature at 900 °C for 10 min, the samples were cooled down to room temperature rapidly, and swiftly inserted in the microscope for TEM inspection.

**Figure 6 materials-16-01291-f006:**
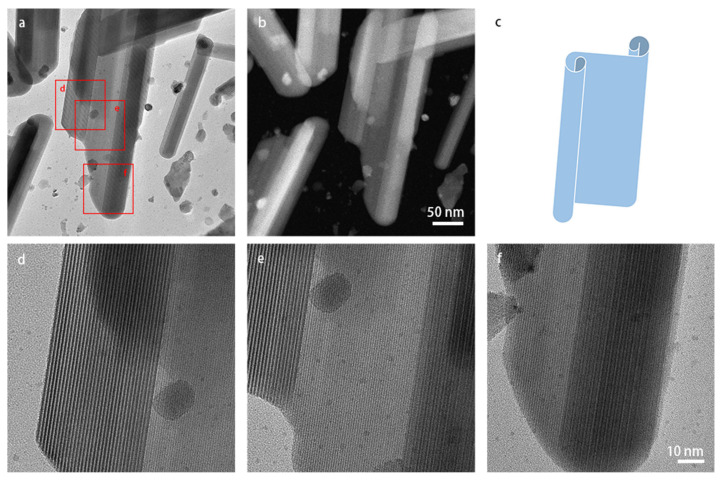
(S)TEM images of a larger lath-shaped 2D nanosheet: (**a**) bright-field TEM image and (**b**) HAADF-STEM image recorded at the same magnification; (**c**) model of the shape of the particles where the edges on the left-hand side and right-hand side are curled up; (**d**–**f**) magnified TEM images of the corresponding areas indicated with red squares in panel (**a**). Moiré patterns show multiple layers at the left-hand and right-hand edges of the particle. Panels (**d**–**f**) are at the same magnification.

**Figure 7 materials-16-01291-f007:**
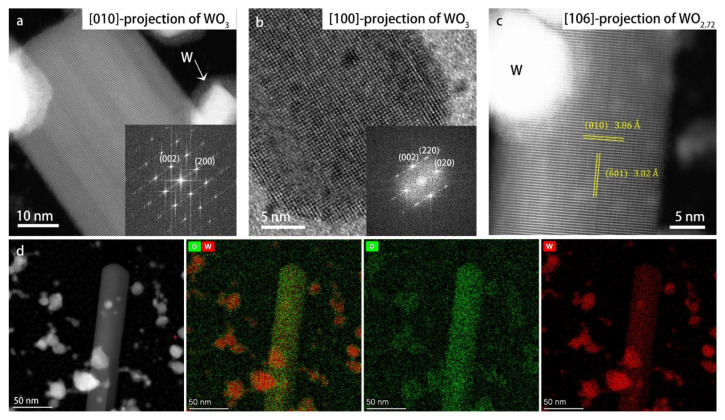
High-resolution (S)TEM images of lath-shaped particles and STEM-EDS chemical mapping results: (**a**,**c**) HR-STEM images; (**b**) HR-TEM images; (**d**) EDS chemical mapping performed in STEM mode. The chemical maps of W (red) and O (green) are shown both separately and overlapping.

**Figure 8 materials-16-01291-f008:**
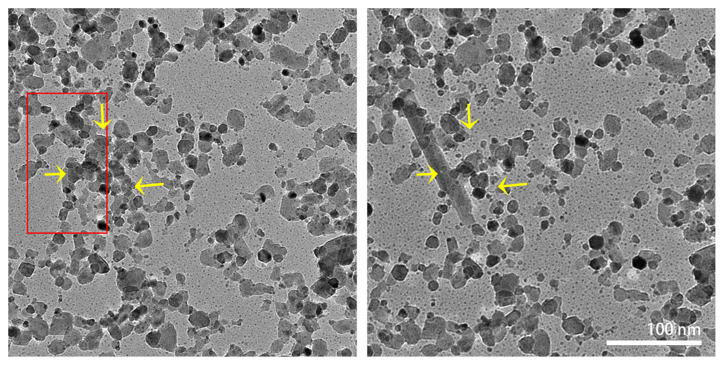
In situ TEM images recorded before (**left**) and after (**right**) the formation of the lath-shaped particle at 900 °C. The position of the lath-shaped particle is marked with a red rectangle. Yellow arrows indicate the positions where the WO_3_ nanoparticles disappeared after the formation of the lath-shaped particle. The two images are at the same magnification.

## Data Availability

The data will be made available upon request.

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
