# Peer review of "Formation Pathways of Lath-Shaped WO3 Nanosheets and Elemental W Nanoparticles from Heating of WO3 Nanocrystals Studied via In Situ TEM"

_materials, 2023, doi:10.3390/ma16031291_

Round 1

Reviewer 1 Report

The work is devoted to in situ investigation of high temperature WO3 transformations employing TEM. Based on the authors findings a new route towards to controlled synthesis of W/WO3 nanomaterials can be elaborated.

After a detailed study of the manuscript, several questions arose for the authors:

1) Did you try to provide the same heating experiments (to 900°C) with WO3 cubic NPs in air ? If, yes, what did you observe? The obtained results might reinforce your discussion part.

2) Finally I did not get what is the transition temperature ? It seems that transformation takes part before 900°C. Why not to give more to TEM grid with big amount of sample at 800°C ? at 700°C ? Such ex situ experiments (with further TEM examination) could bring new information and enrich your discussion. 

3) Are lath-shaped nano-sheets are stable under electron beam ? You should mention it in the text.

4) Taking into account low tolerance of raw WO3 NPs towards to electron beam, why did not you try to reduce e-dose of your beam? For example 120kV ?

5) Missed space in DENSsolutions in experimental part.

The research presentation is clear and logical. The authors provide sufficient amount of information to introduce non-topic readers to the domain of WO3 materials. The employment of in stu TEM is also logical and appropriate to the stated objectives of the work. English grammar and style are also ok (except few misprints). Summing up I consider that the present research deserves to published in Materials.

Reviewer 2 Report

The manuscript reports on thermal behavior of cubic tungsten oxide (VI) upon heating to 900°C. The study was carried out using in situ using transmission electron microscopy, including high-resolution TEM which is quite attractive from the methodological point of view. The results are presented in a logical manner and are of potential interest to the readership of the Materials journal. The article is suitable for publication in the Materials journal with some minor corrections.   1. The manuscript states that when WO3 is heated to 900°C, a pure tungsten phase is formed (Figure 4c), while supplementary materials indicate that an admixture of tungsten oxide is present in some areas with lath-shapped particles (Figure S2). Please provide additional data on the phase composition, for example, the results of X-ray diffraction analysis. Please also estimate the mass fraction of undecomposed WO3 in the final product. 2. Discussion of the thermal behavior of inorganic materials may be incomplete without thermal analysis data. Please provide the relevant TGA/DSC data for cubic tungsten oxide or add a short description of the recently reported data. 3. The manuscript states that the tungsten particles were re-oxidized in air based on the EDX data which were collected several weeks after the W particles were obtained. Whether these data is necessary regarding the scope of the manuscript. Please re-consider or argue. 4. A very interesting point of the manuscript is the mentioning of the unusual high-temperature monoclinic WO3 phase which was discussed by Howard 10.1088/0953-8984/14/3/308 and Locherer 10.1088/0953-8984/11/21/303. As far as I know, the most recent survey on the topic was provided in 10.1007/s10973-020-09345-z. In the manuscript, the information of this unusual phase transition is wirth focusing along with a short survey with the relevant references. 5. The catalog number of WO3 nanoparticles (Sigma-Aldrich) is not specified in the experimental part. Please specify. 6. Please add scale bars in Figures 6 (a, d-e), 8 (left).

Reviewer 3 Report

This manuscript studied phase transformations of cubic WO3 nanoparticles through both in-situ and ex-situ heating, and found that heating of a low concentration of WO3 nanoparticles leads to the exclusive formation of elemental W nanoparticles, while the heating of a high concentration of WO3 nanoparticles lead to the predominant formation of lath-shaped WO3-x nanosheets. A solid-vapor-solid growth mechanism was further proposed to explain the TEM observed phase transformation of cubic WO3.

This manuscript is interesting and well organized. The conclusion is supported by sufficient data presented in the manuscript. But it should be noted there are some minor grammatical mistakes exist, for example, the misuse of definite articles. Therefore, I suggest publishing this manuscript after those noticeable grammatical mistakes have been corrected.

Author Response

Author Reply: We would like to thank the Reviewer for these kind words and the positive assessment. We have checked the English and grammar of the whole text, and made small modifications everywhere. We hope that the Reviewer will be satisfied with the revision of the manuscript.